# Learning dynamic polynomial proofs

**Alhussein Fawzi**
DeepMind
afawzi@google.com

**Mateusz Malinowski**
DeepMind
mateuszm@google.com

**Hamza Fawzi**
University of Cambridge
hf323@cam.ac.uk

**Omar Fawzi**
ENS Lyon
omar.fawzi@ens-lyon.fr

## Abstract

Polynomial inequalities lie at the heart of many mathematical disciplines. In this paper, we consider the fundamental computational task of automatically searching for proofs of polynomial inequalities. We adopt the framework of *semi-algebraic proof systems* that manipulate polynomial inequalities via elementary inference rules that infer new inequalities from the premises. These proof systems are known to be very powerful, but searching for proofs remains a major difficulty. In this work, we introduce a machine learning based method to search for a *dynamic* proof within these proof systems. We propose a deep reinforcement learning framework that learns an embedding of the polynomials and guides the choice of inference rules, taking the inherent symmetries of the problem as an inductive bias. We compare our approach with powerful and widely-studied linear programming hierarchies based on *static* proof systems, and show that our method reduces the size of the linear program by several orders of magnitude while also improving performance. These results hence pave the way towards augmenting powerful and well-studied semi-algebraic proof systems with machine learning guiding strategies for enhancing the expressivity of such proof systems.

## 1 Introduction

Polynomial inequalities abound in mathematics and its applications. Many questions in the areas of control theory [Par00], robotics [MAT13], geometry [PP04], combinatorics [Lov79], program verification [MFK⁺16] can be modeled using polynomial inequalities. For example, deciding the stability of a control system can be reduced to proving the nonnegativity of a polynomial [PP02]. Producing *proofs* of polynomial inequalities is thus of paramount importance for these applications, and has been a very active field of research [Las15].

To produce such proofs, we rely on *semi-algebraic proof systems*, which define a framework for manipulating polynomial inequalities. These proof systems define inference rules that generate new polynomial inequalities from existing ones. For example, inference rules can state that the product and sum of two non-negative polynomials is non-negative. Given a polynomial $f(\mathbf{x})$, a proof of global non-negativity of $f$ consists of a sequence of applications of the inference rules, starting from a set of axioms, until we reach the target statement. Finding such a path is in general a very complex task. To overcome this, a very popular approach in polynomial optimization is to use *hierarchies* that are based on *static* proof systems, whereby inference rules are unrolled for a *fixed* number of steps, and convex optimization is leveraged for the proof search. Despite the great success of such methods in computer science and polynomial optimization [Lau03, CT12], this approach however can suffer from a lack of expressivity for lower levels of the hierarchy, and a curse of dimensionality at higher levels of the hierarchy. Moreover, such static proofs significantly depart from our common

conception of the proof search process, which is inherently *sequential*. This makes static proofs difficult to interpret.

In this paper, we use machine learning to guide the search of a *dynamic proof* of polynomial inequalities. We believe this is the first attempt to use machine learning to search for semi-algebraic proofs. Specifically, we list our main contributions as follows:

- We propose a novel neural network architecture to handle polynomial inequalities with built-in support for the symmetries of the problem.

- Leveraging the proposed architecture, we train a *prover* agent with DQN [MKS+13] in an unsupervised environment; i.e., without having access to any existing proof or ground truth information.

- We illustrate our results on the *maximum stable set problem*, a well known combinatorial problem that is intractable in general. Using a well-known semi-algebraic proof system [LS91, SA90], we show that our dynamic prover significantly outperforms the corresponding static, unrolled, method.

**Related works.** Semi-algebraic proof systems have been studied by various communities *e.g.*, in real algebraic geometry, global optimization, and in theoretical computer science. Completeness results for these proof systems have been obtained in real algebraic geometry, e.g., [Kri64, Ste74]. In global optimization, such proof systems have led to the development of very successful convex relaxations based on static hierarchies [Par00, Las01, Lau03]. In theoretical computer science, static hierarchies have become a standard tool for algorithm design [BS14], often leading to optimal performance. Grigoriev et al. [GHP02] studied the proof complexity of various problems using different semi-algebraic proof systems. This fundamental work has shown that problems admitting proofs of very large static degree can admit a compact dynamic proof. While most previous works has focused on understanding the power of bounded-degree static proofs, there has been very little work on devising strategies to search for *dynamic* proofs, and our work is a first step in this direction.

Recent works have also studied machine learning strategies for automated theorem proving [BLR+19, HDSS18, KUMO18, GKU+18]. Such works generally build on existing theorem provers and seek to improve the choice of inference rules or tactics at each step of the proof. In contrast, our work does not rely on existing theorem provers and instead uses elementary inference rules in the context of semi-algebraic systems. We see these two lines of works as complementary, as building improved provers for polynomial inequalities can provide a crucial tactic that integrates into general ATP systems. We finally note that prior works have applied neural networks to combinatorial optimization problems [BLP18], such as the satisfiability problem [SLB+18]. While such techniques seek to show the *existence* of good-quality feasible points (e.g., a satisfying assignment), we emphasize that we focus here on proving statements *for all* values in a set (e.g., showing the *non*existence of any satisfying assignment) – i.e., $\exists$ vs $\forall$. Finally, we note that the class of polynomial optimization contains combinatorial optimization problems as a special case.

**Notations.** We let $\mathbb{R}[\mathbf{x}]$ denote the ring of multivariate polynomials in $\mathbf{x} = (x_1, \ldots, x_n)$. For $\alpha \in \mathbb{N}^n$ and $\mathbf{x} = (x_1, \ldots, x_n)$, we let $\mathbf{x}^\alpha = x_1^{\alpha_1} \cdots x_n^{\alpha_n}$. The degree of a monomial $\mathbf{x}^\alpha$ is $|\alpha| = \sum_{i=1}^n \alpha_i$. The degree of any polynomial in $\mathbb{R}[\mathbf{x}]$ is the largest degree of any of its monomials. For $n \in \mathbb{N}$, we use $[n]$ to denote the set $\{1, \ldots, n\}$. We use $|\cdot|$ to denote the cardinality of a finite set.

## 2 Problem modeling using polynomials

To illustrate the scope of this paper, we review the connection between optimization problems and proving the non-negativity of polynomials. We also describe the example of the stable set problem, which we will use as a running example throughout the paper.

**Polynomial optimization.** A general polynomial optimization problem takes the form

$$\text{maximize} \quad f(\mathbf{x}) \quad \text{subject to} \quad \mathbf{x} \in \mathcal{S}. \tag{1}$$

where $f(\mathbf{x})$ is a polynomial and $\mathcal{S}$ is a basic closed semi-algebraic set defined using polynomial equations and inequalities $\mathcal{S} = \{\mathbf{x} \in \mathbb{R}^n : g_i(\mathbf{x}) \geq 0, h_j(\mathbf{x}) = 0 \ \forall i, j\}$, where $g_i, h_j$ are arbitrary polynomials. Such problem subsumes many optimization problems as a special case. For example using the polynomial equality constraints $x_i^2 = x_i$ restricts $x_i$ to be an integer in $\{0, 1\}$. As such,

integer programming is a special case of (1). Problem (1) can also model many other optimization problems that arise in theory and practice, see e.g., [Las15].

**Optimization and inequalities.** In this paper we are interested in proving *upper bounds* on the optimal value of (1). Proving an upper bound of $\gamma$ on the optimal value of (1) amounts to proving that

$$\forall \mathbf{x} \in \mathcal{S}, \quad \gamma - f(\mathbf{x}) \geq 0. \tag{2}$$

We are looking at proving such inequalities using semi-algebraic proof systems. Therefore, developing tractable approaches to proving nonnegativity of polynomials on semialgebraic sets has important consequences on polynomial optimization.

**Remark 1.** *We note that proving an upper bound on the value of* (1) *is more challenging than proving a lower bound. Indeed, to prove a lower bound on the value of the maximization problem* (1) *one only needs to exhibit a feasible point* $\mathbf{x}_0 \in \mathcal{S}$; *such a feasible point implies that the optimal value is* $\geq f(\mathbf{x}_0)$. *In contrast, to prove an upper bound we need to prove a polynomial inequality, valid for all* $\mathbf{x} \in \mathcal{S}$ *(notice the* $\forall$ *quantifier in* (2)).

**Stable sets in graphs.** We now give an example of a well-known combinatorial optimization problem, and explain how it can be modeled using polynomials. Let $G = (V, E)$ denote a graph of $n = |V|$ nodes. A *stable set* $S$ in $G$ is a subset of the vertices of $G$ such that for every two vertices in $S$, there is no edge connecting the two. The *stable set* problem is the problem of finding a stable set with largest cardinality in a given graph. This problem can be formulated as a polynomial optimization problem as follows:

$$\begin{array}{ll} \underset{\mathbf{x} \in \mathbb{R}^n}{\text{maximize}} & \sum_{i=1}^n x_i \\ \text{subject to} & x_i x_j = 0 \text{ for all } (i,j) \in E, \\ & x_i^2 = x_i \text{ for all } i \in \{1, \ldots, n\}. \end{array} \tag{3}$$

The constraint $x_i^2 = x_i$ is equivalent to $x_i \in \{0, 1\}$. The variable $\mathbf{x} \in \mathbb{R}^n$ is interpreted as the characteristic function of $S$: $x_i = 1$ if and only if vertex $i$ belongs to the stable set $S$. The cardinality of $S$ is measured by $\sum_{i=1}^n x_i$, and the constraint $x_i x_j = 0$ for $ij \in E$ disallows having two nodes in $S$ that are connected by an edge. Finding a stable set of largest size is a classical NP-hard problem, with many diverse applications [Lov79, Sch03]. As explained earlier for general polynomial optimization problems, showing that there is *no* stable set of size larger than $\gamma$ corresponds to showing that $\gamma - \sum_{i=1}^n x_i \geq 0$ for all $\mathbf{x}$ verifying the constraints of (3).

## 3 Static and dynamic semi-algebraic proofs

A semi-algebraic proof system is defined by elementary *inference rules*, which produce non-negative polynomials. Specifically, a proof consists in applying these inference rules starting from a set of axioms $g_i(\mathbf{x}) \geq 0, h_j(\mathbf{x}) = 0$ until we reach a desired inequality $p \geq 0$.[1]

In this paper, we will focus on proving polynomial inequalities valid on the hypercube $[0,1]^n = \{\mathbf{x} \in \mathbb{R}^n : 0 \leq x_i \leq 1, \ \forall i = 1, \ldots, n\}$. As such, we consider the following inference rules, which appear in the so-called Lovász-Schrijver (LS) proof system [LS91] as well as in the Sherali-Adams framework [SA90]:

$$\frac{g \geq 0}{x_i g \geq 0} \qquad \frac{g \geq 0}{(1 - x_i)g \geq 0} \qquad \frac{g_i \geq 0}{\sum_i \lambda_i g_i \geq 0, \forall \lambda_i \geq 0}, \tag{4}$$

where $\frac{A}{B}$ denotes that $A$ implies $B$. The *proof* of a statement (i.e., non-negativity of a polynomial $p$) consists in the *composition* of these elementary inference rules, which exactly yields the desired polynomial $p$. Starting from the axiom $1 \geq 0$, the composition of inference rules in Eq. (4) yields functions of the form $\sum_{\alpha, \beta} \lambda_{\alpha, \beta} \mathbf{x}^\alpha (1 - \mathbf{x})^\beta$, where $\alpha = (\alpha_1, \ldots, \alpha_n) \in \mathbb{N}^n$ and $\beta = (\beta_1, \ldots, \beta_n) \in \mathbb{N}^n$ are tuples of length $n$, and $\lambda_{\alpha, \beta}$ are non-negative coefficients. It is clear that all polynomials of this form are non-negative for all $\mathbf{x} \in [0,1]^n$, as they consist in a composition of the inference rules (4). As such, writing a polynomial $p$ in this form gives a *proof* of non-negativity of $p$ on the hypercube. The following theorem shows that such a proof always exists provided we assume $p(\mathbf{x})$ is

strictly positive for all $\mathbf{x} \in [0, 1]^n$. In words, this shows that the set of inference rules (4) forms a complete proof system[2]:

**Theorem 1** ([Kri64] Positivstellensatz). *Assume $p$ is a polynomial such that $p(\mathbf{x}) > 0$ for all $\mathbf{x} \in [0, 1]^n$. Then there exists an integer $l$, and nonnegative scalars $\lambda_{\alpha,\beta} \geq 0$ such that*

$$p(\mathbf{x}) = \sum_{|\alpha|+|\beta| \leq l} \lambda_{\alpha,\beta} \mathbf{x}^\alpha (1 - \mathbf{x})^\beta. \tag{5}$$

**Static proofs.** Theorem 1 suggests the following approach to proving non-negativity of a polynomial $p(\mathbf{x})$: fix an integer $l$ and search for non-negative coefficients $\lambda_{\alpha,\beta}$ (for $|\alpha|+|\beta| \leq l$) such that (5) holds. This *static* proof technique is one of the most widely used approaches for finding proofs of polynomial inequalities, as it naturally translates to solving a convex optimization problem [Lau03]. In fact, (5) is a *linear* condition in the unknowns $\lambda_{\alpha,\beta}$, as the functional equality of two polynomials is equivalent to the equality of the coefficients of each monomial. Thus, finding such coefficients is a linear program where the number of variables is equal to the number of tuples $(\alpha, \beta) \in \mathbb{N}^n \times \mathbb{N}^n$ such that $|\alpha| + |\beta| \leq l$, i.e., of order $\Theta(n^l)$ for $l$ constant. The collection of these linear programs gives a *hierarchy*, indexed by $l \in \mathbb{N}$, for proving non-negativity of polynomials. Theorem 1 shows that as long as $p > 0$ on $[0, 1]^n$ there exists $l$ such that $p$ can be proved nonnegative by the $l$'th level of the hierarchy. However, we do not know *a priori* the value of $l$. In fact this value of $l$ can be much larger than the degree of the polynomial $p$. In other words, in order to prove the non-negativity of a low-degree polynomial $p$, one may need to manipulate high-degree polynomial expressions and leverage cancellations in the right-hand side of (5) – see illustration below for an example.

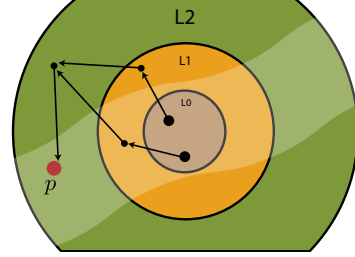

Figure 1: Illustration of a dynamic *vs.* static proof. Each concentric circle depicts the set of polynomials that can be proved non-negative by the $l$'th level of the hierarchy. The wiggly area is the set of polynomials of degree e.g., $1$. A dynamic proof (black arrows) of $p \geq 0$ seeks an (adaptive) sequence of inference rules that goes from the initial set of axioms (dots in $L0$) to target $p$.

**Dynamic proofs.** For large values of $l$, the linear program associated to the $l$'th level of the hierarchy is prohibitively large to solve. To remedy this, we propose to search for *dynamic* proofs of non-negativity. This technique relies on proving intermediate *lemmas* in a sequential way, as a way to find a concise proof of the desired objective. Crucially, the choice of the *intermediate lemmas* is strongly problem-dependent – it depends on the target polynomial $p$, in addition to the axioms and previously derived lemmas. This is in stark contrast with the *static* approach, where hierarchies are problem-independent (e.g., they are obtained by limiting the degree of proof generators, the $\mathbf{x}^\alpha(1 - \mathbf{x})^\beta$ in our case). In spite of the benefits of a dynamic proof system, searching for these proofs is a challenging problem on its own, where one has to decide on inference rules applied at each step of the proof. We also believe such a dynamic proving approach is more aligned with human reasoning, which is also a sequential process where intuition plays an important role in deriving new lemmas by applying suitable inference rules that lead to interpretable proofs. We finally note that the dynamic proving strategy subsumes the static one, as a static proof can be seen as a non-adaptive version of a dynamic proof.

**Illustration.** To illustrate the difference between the static and dynamic proof systems, consider the stable set problem in Sect. 2 on the complete graph on $n$ nodes, where each pair of nodes is connected. It is clear that the maximal stable set has size 1; this can be formulated as follows:[3]

$$\begin{cases} x_i^2 = x_i, \ i = 1, \ldots, n \\ x_i x_j = 0, \ \forall i \neq j \\ x_i \geq 0, 1 - x_i \geq 0, \ i = 1, \ldots, n \end{cases} \Rightarrow \quad 1 - \sum_{i=1}^n x_i \geq 0. \tag{6}$$

In the static framework, we seek to express the polynomial $1 - \sum_{i=1}^n x_i$ as in (5), *modulo* the equalities $x_i x_j = 0$. One can verify that

$$1 - \sum_{i=1}^n x_i \ = \ \prod_{i=1}^n (1 - x_i) \qquad \mod \quad (x_i x_j = 0, \ \forall i \neq j). \tag{7}$$

The proof in Equation (7) is a static proof of degree $n$ because it involves the degree $n$ product $\prod_{i=1}^{n}(1 - x_i)$. This means that the proof (7) will only be found at level $n$ of the static hierarchy, which is a linear program of size exponential in $n$. One can further show that it is necessary to go to level at least $n$ to find a proof of (6) (cf. Supp. Mat).

In contrast, one can provide a dynamic proof of the above where the degree of any intermediate lemma is at most two. To see why, it suffices to multiply the polynomials $1 - x_i$ sequentially, each time eliminating the degree-two terms using the equalities $x_i x_j = 0$ for $i \neq j$. The dynamic proof proceeds as follows (note that no polynomial of degree greater than two is ever formed).

$$1 - x_1 \geq 0 \xrightarrow[1 - x_2 \geq 0]{\text{multiply by}} (1 - x_1)(1 - x_2) \geq 0 \xrightarrow[x_1 x_2 = 0]{\text{reduce using}} 1 - x_1 - x_2 \geq 0$$

$$\xrightarrow[1 - x_3 \geq 0]{\text{multiply by}} (1 - x_1 - x_2)(1 - x_3) \geq 0 \xrightarrow[x_1 x_3 = x_2 x_3 = 0]{\text{reduce using}} 1 - x_1 - x_2 - x_3 \geq 0$$

$$\vdots$$

$$\xrightarrow[1 - x_n \geq 0]{\text{multiply by}} (1 - x_1 - \ldots - x_{n-1})(1 - x_n) \geq 0 \xrightarrow[x_i x_n = 0 \text{ for } i < n]{\text{reduce using}} 1 - x_1 - \ldots - x_n \geq 0.$$

## 4 Learning dynamic proofs of polynomials

### 4.1 Reinforcement learning framework for semi-algebraic proof search

We model the task of finding dynamic proofs as an interaction between the agent and an environment, formalized as a Markov Decision Process (MDP), resulting in a sequence of states, actions and observed rewards. The agent state $s_t$ at time step $t$ is defined through the triplet $(f, \mathcal{M}_t, \mathcal{E}_t)$, where:

- $\mathcal{M}_t$ denotes the *memory* at $t$; *i.e.,* the set of polynomials that are known to be non-negative at $t$. This contains the set of polynomials that are *assumed* to be non-negative (i.e., *axioms* $g_i$), as well as intermediate steps (i.e., *lemmas*), which are derived from the axioms through inference rules,
- $\mathcal{E}_t$ denotes the set of *equalities*; *i.e.,* the set of polynomials identically equal to zero,
- $f$ denotes the objective polynomial to bound (cf Section 2).

At each time $t$, the agent selects an action $a_t$ from a set of legal actions $\mathcal{A}_t$, obtained by applying one or more inference rules in Eq. (4) to elements in $\mathcal{M}_t$.[4] Observe that since elements in $\mathcal{M}_t$ are non-negative, the polynomials in $\mathcal{A}_t$ are also non-negative. The selected action $a_t \in \mathcal{A}_t$ is then appended to the memory $\mathcal{M}_{t+1}$ at the next time step. After selecting $a_t$, a reward $r_t$ is observed, indicating how close the agent is to finding the proof of the statement, with higher rewards indicating that the agent is "closer" to finding a proof – see Sect. 4.2 for more details.

The goal of the agent is to select actions that maximize future returns $R_t = \mathbb{E}[\sum_{t'=t}^{T} \gamma^{t'-t} r_{t'}]$, where $T$ indicates the length of an episode, and $\gamma$ is the discount factor. We use a deep reinforcement learning algorithm where the action-value function is modeled using a deep neural network $q_\theta(s, a)$. Specifically, the neural network takes as input a state-action pair, and outputs an estimate of the return; we use the DQN [MKS+13] algorithm for training, which leverages a replay memory buffer for increased stability [Lin92]. We refer to [MKS+13, Algorithm 1] for more details about this approach.

Note that in contrast to many RL scenarios, the action space here grows with $t$, as larger memories mean that more lemmas can be derived. The large action space makes the task of finding a dynamic proof particularly challenging; we therefore rely on dense rewards (Sect. 4.2) and specialized architectures (Sect. 4.3) for tackling this problem.

### 4.2 Reward signal

We now describe the reward signal $r_t$. One potential choice is to assign a positive reward ($r_t > 0$) when the objective $\gamma^* \geq f$ is reached (where $\gamma^*$ is the optimal bound) and zero otherwise. However, this suffers from two important problems: 1) the reward is *sparse*, which makes learning difficult, 2) this requires the knowledge of the optimal bound $\gamma^*$. Here, we rely instead on a *dense* and

*unsupervised* reward scheme, where positive reward is given whenever the chosen action results in an improvement of the bound.

More formally, at each step $t$, we solve the following linear program:

$$\min_{\gamma_t, \{\lambda\}} \gamma_t \quad \text{subject to} \quad \gamma_t - f = \sum_{i=1}^{|\mathcal{M}_t|} \lambda_i m_i, \quad \lambda \geq 0, \tag{8}$$

where $\{m_i\}$ denote the polynomials in $\mathcal{M}_t$. Note that the constraint in Eq. (8) is a *functional* equality of two polynomials, which is equivalent to the equality of the coefficients of the polynomials. In words, Eq. (8) computes the optimal upper bound $\gamma_t$ on $f$ that can be derived through a non-negative linear combination of elements in the memory; in fact, since $\sum_{i=1}^{|\mathcal{M}_t|} \lambda_i m_i$ is non-negative, we have $f \leq \gamma_t$. Crucially, the computation of the bound in Eq. (8) can be done very efficiently, as $\mathcal{M}_t$ is kept of small size in practice (e.g., $|\mathcal{M}_t| \leq 200$ in the experiments).

Then, we compute the reward as the relative improvement of the bound: $r_t = \gamma_{t+1} - \gamma_t$, where $r_t$ is the reward observed after taking action $a_t$. Note that positive reward is observed only when the chosen action $a_t$ leads to an improvement of the current bound. We emphasize that this reward attribution scheme alleviates the need for any supervision during our training procedure; specifically, the agent does not require human proofs or even estimates of bounds for training.

## 4.3 Q-network with symmetries

The basic objects we manipulate are polynomials and sets of polynomials, which impose natural symmetry requirements. We now describe how we build in symmetries in our Q-network $q_\theta$.

Our Q-network $q_\theta$, takes as input the state $s_t = (f, \mathcal{M}_t, \mathcal{E}_t)$, as well as the action polynomial $a_t$. We represent polynomials as vectors of coefficients of size $N$, where $N$ is the number of possible monomials. While sets of polynomials (e.g., $\mathcal{M}_t$) can be encoded with a matrix of size $c \times N$, where $c$ denotes the cardinality of the set, such an encoding does not take into account the *orderless* nature of sets. We, therefore, impose our Q-value function to be invariant to the order of enumeration of elements in $\mathcal{M}$, and $\mathcal{E}$; that is, we require that the following hold for any permutations $\pi$ and $\chi$:

**Symmetry I (orderless sets).** $\quad q_\theta \left( \{m_i\}_{i=1}^{|\mathcal{M}_t|}, \{e_j\}_{j=1}^{|\mathcal{E}_t|}, f, a \right) = q_\theta \left( \{m_{\pi(i)}\}_{i=1}^{|\mathcal{M}_t|}, \{e_{\chi(j)}\}_{j=1}^{|\mathcal{E}_t|}, f, a \right).$

To satisfy the above symmetry, we consider value functions of the form:

$$q_\theta \left( \{m_i\}_{i=1}^{|\mathcal{M}_t|}, \{e_j\}_{j=1}^{|\mathcal{E}_t|}, f, a \right) = \zeta_{\theta^{(3)}} \left( \sigma(V), \sigma(W) \right),$$

where $V = \{v_{\theta^{(1)}}(m_i, f, a)\}_{i=1}^{|\mathcal{M}_t|}, W = \{v_{\theta^{(2)}}(e_j, f, a)\}_{j=1}^{|\mathcal{E}_t|}, v_{\theta^{(1)}}$ and $v_{\theta^{(2)}}$ are trainable neural networks with additional symmetry constraints (see below), $\sigma$ is a symmetric function of the arguments (e.g., max, sum), and $\zeta_{\theta^{(3)}}$ is a trainable neural network.

In addition to the above symmetry, $v_\theta$ has to be well chosen in order to guarantee invariance under relabeling of variables (that is, $x_i \to x_{\pi(i)}$ for any permutation $\pi$). In fact, the variable names do not have any specific meaning *per se*; relabeling all polynomials in the same way results in the exact same problem. We therefore require that the following constraint is satisfied for any permutation $\pi$:

**Symmetry II (variable relabeling).** $\quad v_\theta(m, f, a) = v_\theta(\pi m, \pi f, \pi a), \tag{9}$

where $\pi m$ indicates a permutation of the variables in $m$ using $\pi$. For example, if $\pi$ is such that $\pi(1) = 2, \pi(2) = 3$ and $\pi(3) = 1$, and $m = x_1 + 2x_1 x_3$ then $\pi m = x_2 + 2x_1 x_2$. Note that in the above constraint, the *same* permutation $\pi$ is acting on $m$, $f$ and $a$.

We now describe how we impose this symmetry. Given two triplets of monomials $(\mathbf{x}^{\alpha_1}, \mathbf{x}^{\alpha_2}, \mathbf{x}^{\alpha_3})$ and $(\mathbf{x}^{\beta_1}, \mathbf{x}^{\beta_2}, \mathbf{x}^{\beta_3})$, we say that these two triplets are equivalent (denoted by the symbol $\sim$) iff there exists a permutation $\pi$ such that $\beta_i = \pi(\alpha_i)$ for $i = 1, 2, 3$. For example, $(x_1 x_2, x_2^2, x_2 x_3) \sim (x_1 x_3, x_3^2, x_2 x_3)$. The *equivalence class* $[(\mathbf{x}^{\alpha_1}, \mathbf{x}^{\alpha_2}, \mathbf{x}^{\alpha_3})]$ regroups all triplets of monomials that are equivalent to $(\mathbf{x}^{\alpha_1}, \mathbf{x}^{\alpha_2}, \mathbf{x}^{\alpha_3})$. We denote by $\mathscr{E}$ the set of all such equivalence classes. Our first step to construct $v_\theta$ consists in mapping the triplet $(m, f, a)$ to a feature vector which respects the variable relabeling symmetry. To do so, let $m, f, a$ be polynomials in $\mathbb{R}[\mathbf{x}]$; we consider a feature function that is *trilinear* in $(m, f, a)$; that is, it is linear in each argument $m$, $f$ and $a$. For such a

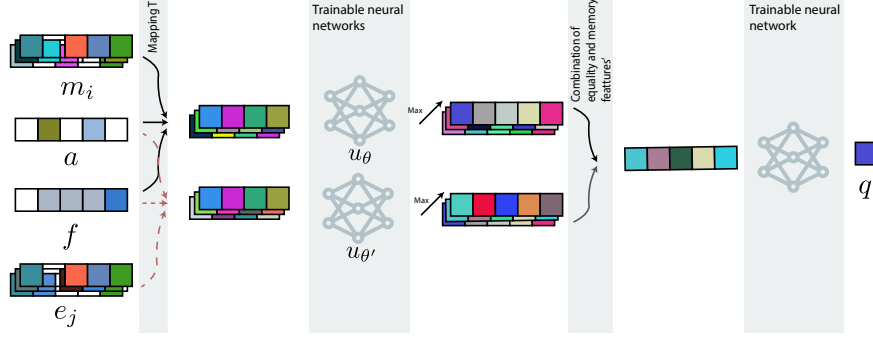

Figure 2: Structure of Q-network. $\{m_i\}$ denotes the set of *axioms* and *lemmas*, $a$ denotes the action, $f$ is the objective function, and $e_j$ denotes the set of equality polynomials.

function, $T : \mathbb{R}[\mathbf{x}] \times \mathbb{R}[\mathbf{x}] \times \mathbb{R}[\mathbf{x}] \to \mathbb{R}^s$ (where $s$ denotes the feature size), we have: $T(m, f, a) = \sum_{\alpha,\beta,\gamma} m_\alpha f_\beta a_\gamma T(\mathbf{x}^\alpha, \mathbf{x}^\beta, \mathbf{x}^\gamma)$. If $(\mathbf{x}^\alpha, \mathbf{x}^\beta, \mathbf{x}^\gamma) \sim (\mathbf{x}^{\alpha'}, \mathbf{x}^{\beta'}, \mathbf{x}^{\gamma'})$, then we set $T(\mathbf{x}^\alpha, \mathbf{x}^\beta, \mathbf{x}^\gamma) = T(\mathbf{x}^{\alpha'}, \mathbf{x}^{\beta'}, \mathbf{x}^{\gamma'})$. In other words, the function $T$ has to be constant on each equivalence class. Such a $T$ will satisfy our symmetry constraint that $T(m, f, a) = T(\pi m, \pi f, \pi a)$ for any permutation $\pi$. For example, the above equality constrains $T(1, x_1, x_1) = T(1, x_i, x_i)$ for all $i$ since $(1, x_1, x_1) \sim (1, x_i, x_i)$, and $T(x_1, x_2, x_3) = T(x_i, x_j, x_k)$ for $i \neq j \neq k$ as $(x_1, x_2, x_3) \sim (x_i, x_j, x_k)$. Note, however, that $T(1, x_1, x_1) \neq T(1, x_i, x_j)$ for $i \neq j$; in fact, $(1, x_1, x_1) \not\sim (1, x_i, x_j)$. Finally, we set $v_\theta = u_\theta \circ T$ where $u_\theta$ is a trainable neural network. Fig. 2 summarizes the architecture we use for the Q-network. We refer to Supp. Mat. for more details about architectures and practical implementation.

## 5 Experimental results

We illustrate our dynamic proving approach on the stable set problem described in Section 2. This problem has been extensively studied in the polynomial optimization literature [Lau03]. We evaluate our method against standard linear programming hierarchies considered in this field. The largest stable set in a graph $G$ is denoted $\alpha(G)$.

**Training setup.** We train our prover on randomly generated graphs of size $n = 25$, where an edge between nodes $i$ and $j$ is created with probability $p \in [0.5, 1]$. We seek dynamic proofs using the proof system in Eq. (4), starting from the axioms $\{x_i \geq 0, 1 - x_i \geq 0, i = 1, \ldots, n\}$ and the polynomial equalities $x_i x_j = 0$ for all edges $ij$ in the graph and $x_i^2 = x_i$ for all nodes $i$. We restrict the number of steps in the dynamic proof to be at most 100 steps and limit the degree of any intermediate lemma to 2. We note that our training procedure is unsupervised and does not require prior proofs, or knowledge of $\alpha(G)$ for learning. We use the DQN approach presented in Sect. 4 and provide additional details about hyperparameters and architecture choices in the Supp. Mat.

We compare our approach to the following static hierarchy of linear programs indexed by $l$:

$$\text{min. } \gamma \quad \text{s.t.} \quad \gamma - \sum_{i=1}^n x_i = \sum_{|\alpha|+|\beta| \leq l} \lambda_{\alpha,\beta} \mathbf{x}^\alpha (1 - \mathbf{x})^\beta \quad \text{mod} \quad \begin{pmatrix} x_i x_j = 0, \ ij \in E \\ x_i^2 = x_i, \ i \in V \end{pmatrix}. \quad (10)$$

This hierarchy corresponds to the level $l$ of the Sherali-Adams hierarchy applied to the maximum stable set problem [LS14, Section 4], which is one of the most widely studied hierarchies for combinatorial optimization [Lau03]. Observe that the linear program (10) has $\Theta(n^l)$ variables and constraints for $l$ constant. By completeness of the hierarchy, we know that solving the linear program (10) at level $l = n$ yields the exact value $\alpha(G)$ of the maximum stable set.

**Results.** Table 1 shows the results of the proposed *dynamic* prover on a test set consisting of random graphs of different sizes.[5] We compare the value obtained by the dynamic prover with a random prover taking random legal actions (from the considered proof system), as well as with the Sherali-Adams hierarchy (10). The reported values correspond to an average over a set of 100 randomly

| $n$ | Dyn. (deg. 2) | Static hierarchy | | | | Random | Size of LP | |
|---|---|---|---|---|---|---|---|---|
| | | $l=2$ | $l=3$ | $l=4$ | $l=5$ | | Dyn. | Static $l=5$ |
| 15 | **3.43** | 7.50 | 5.01 | 3.94 | 3.48 | 5.91 | **130** | $5.9 \times 10^3$ |
| 20 | **3.96** | 10.0 | 6.67 | 5.04 | 4.32 | 8.91 | **140** | $2.6 \times 10^4$ |
| 25 | **4.64** | 12.50 | 8.33 | 6.26 | 5.08 | 12.7 | **150** | $7.6 \times 10^4$ |
| 30 | **5.44** | 15.0 | 10.0 | 7.50 | 6.03 | 15.6 | **160** | $1.9 \times 10^5$ |
| 35 | **6.37** | 17.5 | 11.67 | 8.75 | 7.02 | 19.6 | **170** | $4.2 \times 10^5$ |
| 40 | **7.23** | 20.0 | 13.33 | 10.0 | 8.00 | 23.5 | **180** | $8.3 \times 10^5$ |
| 45 | **8.14** | 22.5 | 15.0 | 11.25 | 9.00 | 28.1 | **190** | $1.5 \times 10^6$ |
| 50 | **8.89** | 25.0 | 16.67 | 12.50 | 10.0 | 31.6 | **200** | $2.6 \times 10^6$ |

Table 1: Evaluation of different methods on 100 randomly sampled problems on the maximal stable set problem. For each method, the average estimated bound is displayed (lower values correspond to better – i.e., tighter – bounds). Moreover, the average size of the linear program in which the proof is sought is reported in the last two columns. The proof size is limited to 100 for the dynamic proof, leading to an LP of size $100 + 2n$, as the problem has $2n$ inequality axioms ($x_i \geq 0, 1 - x_i \geq 0$). Note that the static linear program at level $l$ cannot give a bound smaller than $n/l$; we prove this result in Theorem 1 in Supp. Mat.

**Proof that** $3 - \sum_{i=1}^{7} x_i \geq 0$**:**

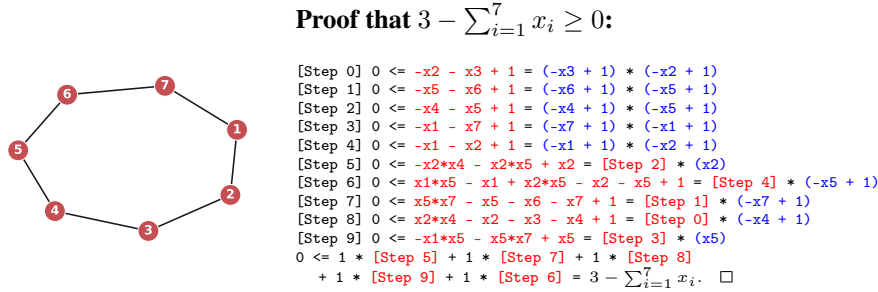

```
[Step 0]  0 <= -x2 - x3 + 1 = (-x3 + 1) * (-x2 + 1)
[Step 1]  0 <= -x5 - x6 + 1 = (-x6 + 1) * (-x5 + 1)
[Step 2]  0 <= -x4 - x5 + 1 = (-x4 + 1) * (-x5 + 1)
[Step 3]  0 <= -x1 - x7 + 1 = (-x7 + 1) * (-x1 + 1)
[Step 4]  0 <= -x1 - x2 + 1 = (-x1 + 1) * (-x2 + 1)
[Step 5]  0 <= -x2*x4 - x2*x5 + x2 = [Step 2] * (x2)
[Step 6]  0 <= x1*x5 - x1 + x2*x5 - x2 - x5 + 1 = [Step 4] * (-x5 + 1)
[Step 7]  0 <= x5*x7 - x5 - x6 - x7 + 1 = [Step 1] * (-x7 + 1)
[Step 8]  0 <= x2*x4 - x2 - x3 - x4 + 1 = [Step 0] * (-x4 + 1)
[Step 9]  0 <= -x1*x5 - x5*x7 + x5 = [Step 3] * (x5)
0 <= 1 * [Step 5] + 1 * [Step 7] + 1 * [Step 8]
   + 1 * [Step 9] + 1 * [Step 6] = 3 - ∑⁷ᵢ₌₁ xᵢ.  □
```

Table 2: An example of proof generated by our agent. Axioms are shown in blue, and derived polynomials (i.e., intermediate lemmas) are shown in red. Note that coefficients in the proof are all *rational*, leading to an exact and fully verifiable proof. See more examples of proofs in the Supp. Mat.

generated graphs. We note that for all methods, bounds are accompanied with a *formal, verifiable, proof*, and are hence correct by definition.

Our dynamic polynomial prover is able to prove an upper bound on $\alpha(G)$ that is better than the one obtained by the Sherali-Adams hierarchy with a linear program that is smaller by several orders of magnitude. For example on graphs of 50 nodes, the Sherali-Adams linear program at level $l = 5$ has more than two million variables, and gives an upper bound on $\alpha(G)$ that is worse than our approach which only uses a linear program of size 200. This highlights the huge benefits that dynamic proofs can offer, in comparison to hierarchy-based static approaches. We also see that our agent is able to learn useful strategies for proving polynomial inequalities, as it significantly outperforms the random agent. We emphasize that while the proposed agent is only trained on graphs of size $n = 25$, it still outperforms all other methods for larger values of $n$ showing good out-of-distribution generalization. Note finally that the proposed architecture which incorporates symmetries (as described in Sect. 4.3) significantly outperforms other generic architectures, as shown in the Supp. Mat.

Table 2 provides an example of a proof produced by our automatic prover, showing that the largest stable set in the cycle graph on 7 nodes is at most 3. Despite the symmetric nature of the graph (unlike random graphs in the training set), our proposed approach leads to human interpretable, and relatively concise proofs. In contrast, the static approach involves searching for a proof in a very large algebraic set.

## 6   Conclusion

Existing hierarchies for polynomial optimization currently rely on a static viewpoint of algebraic proofs and leverage the convexity of the search problem. We propose here a new approach for searching for a dynamic proof using machine learning based strategies. The framework we propose for proving inequalities on polynomials leads to more *natural*, *interpretable* proofs, and significantly outperforms static proof techniques. We believe that augmenting polynomial systems with ML-guided dynamic proofs will have significant impact in application areas such as control theory, robotics, verification, where many problems can be cast as proving polynomial inequalities. One very promising avenue for future research is to extend our dynamic proof search method to other more powerful semi-algebraic proof systems; e.g., based on semi-definite programming.

## Footnotes

[1] In the setting discussed in Section 2, the desired inequality is $p = \gamma - f \geq 0$, where $f$ is the objective function of the optimization problem in (1).

[2]The result is only true for strictly positive polynomials. More precisely, the proof system in (4) is only refutationally complete.

[3]Note that redundant inequalities $x_i \geq 0$ and $1 - x_i \geq 0$ have been added for sake of clarity in what follows.

[4]In practice, we limit ourselves to the first two inference rules (i.e., multiplication by $x_i$ and $1 - x_i$), and find linear combinations using the LP strategy described in Section 4.2. This yields action spaces $\mathcal{A}_t$ of size $2n|\mathcal{M}_t|$.

[5]Despite training the network on graphs of fixed size, we can test it on graphs of any size, as the embedding dimension is independent of $n$. In fact, it is equal to the number of equivalence classes $|\mathscr{E}|$.

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
