[Supplementary Material]

# Learning dynamic polynomial proofs
# (Supplementary material)

**Alhussein Fawzi**
DeepMind
afawzi@google.com

**Mateusz Malinowski**
DeepMind
mateuszm@google.com

**Hamza Fawzi**
University of Cambridge
hf323@cam.ac.uk

**Omar Fawzi**
ENS Lyon
omar.fawzi@ens-lyon.fr

## A   Proofs

### A.1   Static hierarchy for stable set.

We first prove that the static proof on the stable set problem cannot achieve a better bound than $n/l$, a result that is highlighted in the experimental section of the main paper.

**Theorem 1.** *Let $\gamma_l$ be the value of the linear program in Equation (10) of the main paper; that is,*

$$\gamma_l = \textit{min. } \gamma \ \textit{ s.t.} \quad \gamma - \sum_{i=1}^{n} x_i = \sum_{|\alpha+\beta| \leq l} \lambda_{\alpha,\beta} \mathbf{x}^{\alpha}(1-\mathbf{x})^{\beta} \mod \left( \begin{array}{l} x_i x_j = 0, \ ij \in E \\ x_i^2 = x_i, \ i \in V \end{array} \right). \quad (1)$$

*Then $\gamma_l \geq n/l$.*

*Proof.* Assume we have the expression

$$\gamma - \sum_{i=1}^{n} x_i = \sum_{|\alpha+\beta| \leq l} \lambda_{\alpha,\beta} x^{\alpha}(1-x)^{\beta} \mod \left( \begin{array}{l} x_i x_j = 0, \ ij \in E \\ x_i^2 = x_i, \ i \in V \end{array} \right) \quad (2)$$

where the coefficients $\lambda_{\alpha,\beta}$ are nonnegative. We will show that $\gamma \geq n/l$.

- The constant coefficient on the right-hand side of (2) is

$$\sum_{\beta} \lambda_{\emptyset,\beta}. \quad (3)$$

- For $i \in \{1, \ldots, n\}$, the coefficient of $x_i$ in the expression on the right-hand side of (2) is

$$-\sum_{\beta:i\in\beta} \lambda_{\emptyset,\beta} + \sum_{\beta:i\notin\beta} \lambda_{i,\beta} \quad (4)$$

  The presence of the first term is clear and comes from the fact that $(1-x)^{\beta} = 1 - \sum_{i\in\beta} x_i +$ (terms of degree at least two). For the second term in (4) note that

$$x_i(1-x)^{\beta} = x_i - x_i \sum_{j\in\beta} x_j + (\text{terms of degree at least 3}) \quad (5)$$

  If $i \in \beta$ then the term $x_i$ on the RHS of (5) will be cancelled by $x_i^2 = x_i$. This explains why in the second summation in (4) we have to take only the $\beta$'s such that $i \notin \beta$.

Equating coefficients we see that we must have:

$$\begin{cases} \gamma & = \sum_{|\beta| \leq l} \lambda_{\emptyset,\beta} \\ 1 & = \sum_{\beta:i\in\beta} \lambda_{\emptyset,\beta} - \sum_{\beta:i\notin\beta} \lambda_{i,\beta} \qquad \forall i \in \{1,\dots,n\}. \end{cases} \tag{6}$$

Since the coefficients $\lambda_{\alpha,\beta}$ are nonnegative, the second line of (6) tells us that $\sum_{\beta:i\in\beta} \lambda_{\emptyset,\beta} \geq 1$ for all $i \in \{1,\dots,n\}$. We thus get that

$$\gamma = \sum_{|\beta|\leq l} \lambda_{\emptyset,\beta} \geq \frac{1}{l} \sum_{i=1}^{n} \sum_{|\beta|\leq l, i\in\beta} \lambda_{\emptyset,\beta} \geq \frac{n}{l}$$

as desired. $\qquad\qquad\qquad\qquad\qquad\qquad\qquad\qquad\qquad\qquad\qquad\qquad\qquad\qquad\qquad\qquad\square$

Note that when applying Theorem 1 to the complete graph (for which $\alpha(G) = 1$), we see that the $n$th level of the hierarchy is *necessary* in order to obtain an optimal bound of 1.

# B  Implementation details

We now provide implementation details regarding the proposed prover.

## B.1  Computing $T$

Our architecture relies on the computation of the trilinear mapping $T$, which maps triplets of polynomials to feature vectors in $\mathbb{R}^s$, where $s$ denotes a user-specified feature size. Note that $T$ can be re-written as follows

$$T(m, f, a) = \sum_{\alpha,\beta,\gamma} m_\alpha f_\beta a_\gamma T(\mathbf{x}^\alpha, \mathbf{x}^\beta, \mathbf{x}^\gamma) \tag{7}$$

$$= \sum_{[(\mathbf{x}^\alpha, \mathbf{x}^\beta, \mathbf{x}^\gamma)] \in \mathscr{E}} T(\mathbf{x}^\alpha, \mathbf{x}^\beta, \mathbf{x}^\gamma) \sum_{(\mathbf{x}^{\alpha'}, \mathbf{x}^{\beta'}, \mathbf{x}^{\gamma'}) \in [(\mathbf{x}^\alpha, \mathbf{x}^\beta, \mathbf{x}^\gamma)]} m_{\alpha'} f_{\beta'} a_{\gamma'}, \tag{8}$$

where we used the property that $T$ is the same for elements of the same equivalence class. We represent the mapping $T$ in practice with a matrix $\mathbf{T}$ of size $s \times |\mathscr{E}|$ (with each column equal to $T(\mathbf{x}^\alpha, \mathbf{x}^\beta, \mathbf{x}^\gamma)$, for the different equivalence classes in $\mathscr{E}$). In practice, we compute the tensor product $m \otimes f \otimes a \in \mathbb{R}^{N \times N \times N}$ (where each polynomial $m, f, a$ is represented as a vector of size the number of monomials $N$); Eq. (8) then corresponds to the matrix vector multiplication between the matrix $\mathbf{T}$ and the vector $\mathbf{z} \in \mathbb{R}^{|\mathscr{E}|}$, where $z_i = \sum_{(\alpha,\beta,\gamma)\in e_i} [m \otimes f \otimes a]_{\alpha,\beta,\gamma}$, and $e_i$ denotes the $i$-th equivalence class.

## B.2  Architecture details

We now describe in more detail the specifics of the architecture used for our Q-network. Recall from Section 4 in the main paper that we consider a Q-network of the form

$$q_\theta \left( \{m_i\}_{i=1}^{|\mathcal{M}_t|}, \{e_j\}_{j=1}^{|\mathcal{E}_t|}, f, a \right) = \zeta \left( \sigma(V), \sigma(W) \right),$$

where $V = \{v_{\theta^{(1)}}(m_i, f, a)\}_{i=1}^{|\mathcal{M}_t|}$, $W = \{v_{\theta^{(2)}}(e_j, f, a)\}_{j=1}^{|\mathcal{E}_t|}$. To satisfy the re-labeling symmetry, recall that we set $v_\theta = u_\theta \circ T$. In practice, we set $u_{\theta^{(i)}}$ to be a two-layer fully connected neural network with ReLU non-linearity. We set the size of the intermediate layers and output layer to 500. The $\sigma$ operator is set to $\max$. Moreover, the function $\zeta$ is built in two-steps; we first take a $\max$ operation to aggregate the features from equalities and memory elements (and hence obtain a feature vector of size 500), followed by a two-layer fully connected neural network with ReLU non-linearities. The size of the intermediate features is also set to 500, while the output of the neural network is a scalar representing the $q$ value. Note that we have experimented with larger feature sizes/deeper networks, and we did not observe significant improvements of this choice.

We illustrate in Fig. 1 the learning curves obtained with the proposed architecture, as well as a baseline architecture which does *not* consider built-in symmetries for variable relabeling. Specifically, the mapping $T$ is replaced with a linear mapping of the vector resulting from the concatenation of the input polynomials. We see that the proposed architecture leads to significantly better bound compared to the baseline vanilla architecture.

Figure 1: Learning curves showing the average estimated upper bound on a validation set vs. number of enivronment steps. The learning curve of the *proposed* architecture (which incorporates symmetries) is shown in blue. To show the importance of the built-in symmetries, we compare it to a vanilla architecture (red), where $T$ is chosen to be a standard *linear* mapping on the vector of concatenated polynomials in the input (rather than the trilinear mapping described in Eq. (7)). Illustration on the stable set problem with $n = 20$.

## B.3 Leveraging the structure of the Q-network for faster training

For training the prover agent, we use the DQN algorithm in [MKS$^+$13, Algorithm 1]. In Q-Learning, the state-action value function needs to be computed for every action, as the $\epsilon$-greedy strategy chooses the action with largest $q$-value with probability $1 - \epsilon$. To accelerate the computation of the $q$ value for all actions, we leverage the structure of our Q-network. Specifically, at time step $t$, we distinguish between two classes of actions: 1) *new actions*, which are made possible due to the the newly derived inequality $a_t$; e.g., $a_t x_i$, or $a_t(1 - x_i)$ 2) *old actions*, which do not depend on the new element added to the memory. While we do need to compute the $q$ values for all *new actions*, observe that we can re-use computations from previous time steps to speed up the computation of $q(s_{t+1}, a)$ where $a$ is an old action. Specifically, we note that the only difference between $s_t$ and $s_{t+1}$ is that $a_t$ has been added to the memory; that is, $\mathcal{M}_{t+1} = \mathcal{M}_t \cup \{a_t\}$. Hence, observe that

$$\max_{m \in \mathcal{M}_{t+1}} v_{\theta^{(1)}}(m, f, a) = \max\left(F_t, v_{\theta^{(1)}}(a_t, f, a)\right),$$

where $F_t = \max_{m \in \mathcal{M}_t}(v_{\theta^{(1)}}(m, f, a))$. By caching $F_t$, we can therefore significantly speed up the computation of the $q$-value for old actions, which allows us to scale to larger problem instances. This leads to significant speed-ups in our setting where the number of actions is large; specifically, we have $|\mathcal{A}_t| = 2n|\mathcal{M}_t|$.

## B.4 Hyperparameters

We now describe the hyperparameters we use for learning our prover agent. We use RMSProp optimizer with a learning rate $10^{-5}$, and train our model for $1e6$ steps. We set the size of the replay memory to 100, and the degree of any intermediate polynomial to 2. We use a batch size of 32, discount factor of 0.99, $\epsilon = 0.1$, an $\ell_1$ loss for the TD-error [SB18], and initialize the weights of the network with one fixed seed (0). The training is performed on a single GPU (NVIDIA V100).

## C Complementary experimental results

Table 1 shows more examples of proofs. Fig. 2 shows a comparison between the dynamic and static approaches in terms of the obtained bound and the size of the LP for 100 instances of random graphs of size $n = 50$.

Figure 2: Comparison of dynamic and static results on $100$ problem instances of size $n = 50$ nodes. A problem instance is represented by a point in the plane, where the $y$ axis denotes the difference between the static bound and the dynamic bound (i.e., $\gamma_{\text{static}} - \gamma_{\text{dynamic}}$), and the $x$ axis represents the number of proof generators involved in the LP. The dashed line represents the size of the LP when using our dynamic approach (which is limited to $100$, excluding the axioms). Note that static proofs require a very large number of polynomials in the LP proof search. In contrast, our dynamic approach allows to reach better results with $4 - 5$ orders of magnitude less polynomials. This is despite training our prover on graphs of significantly smaller size ($25$ nodes).

**Proof that** $1 - \sum_{i=1}^{7} x_i \geq 0$**:**

```
[Step 0]  0 <= -x3 - x7 + 1 = (-x3 + 1) * (-x7 + 1)
[Step 1]  0 <= -x3 - x6 - x7 + 1 = [Step 0] * (-x6 + 1)
[Step 2]  0 <= -x3 - x4 - x6 - x7 + 1 = [Step 1] * (-x4 + 1)
[Step 3]  0 <= -x3 - x4 - x5 - x6 - x7 + 1 = [Step 2] * (-x5 + 1)
[Step 4]  0 <= -x2 - x3 - x4 - x5 - x6 - x7 + 1 = [Step 3] * (-x2 + 1)
[Step 5]  0 <= -x1 - x2 - x3 - x4 - x5 - x6 - x7 + 1 = [Step 4] * (-x1 + 1)
0 <= 1 * [Step 5] = 1 - \sum_{i=1}^{7} x_i.  □
```

**Proof that** $5 - \sum_{i=1}^{10} x_i \geq 0$**:**

```
[Step 0]  0 <= -x1 - x3 + 1 = (-x3 + 1) * (-x1 + 1)
[Step 1]  0 <= -x4 - x6 + 1 = (-x6 + 1) * (-x4 + 1)
[Step 2]  0 <= -x10 - x3 + 1 = (-x3 + 1) * (-x10 + 1)
[Step 3]  0 <= -x5 - x7 + 1 = (-x7 + 1) * (-x5 + 1)
[Step 4]  0 <= -x2 - x8 + 1 = (-x8 + 1) * (-x2 + 1)
[Step 5]  0 <= -x1 - x3 - x9 + 1 = [Step 0] * (-x9 + 1)
0 <= 1 * [Step 5] + 1 * (x3) + 1 * [Step 2]
    + 1 * [Step 4] + 1 * [Step 1] + 1 * [Step 3] = 5 - \sum_{i=1}^{10} x_i.  □
```

**Proof that** $4 - \sum_{i=1}^{10} x_i \geq 0$**:**

```
[Step 0]  0 <= -x6 - x8 + 1 = (-x6 + 1) * (-x8 + 1)
[Step 1]  0 <= -x6 - x9 + 1 = (-x6 + 1) * (-x9 + 1)
[Step 2]  0 <= -x4 - x9 + 1 = (-x9 + 1) * (-x4 + 1)
[Step 3]  0 <= -x10 - x7 + 1 = (-x10 + 1) * (-x7 + 1)
[Step 4]  0 <= -x2 - x3 + 1 = (-x3 + 1) * (-x2 + 1)
[Step 5]  0 <= -x4 - x5 + 1 = (-x4 + 1) * (-x5 + 1)
[Step 6]  0 <= x10*x9 - x10 - x7 - x9 + 1 = [Step 3] * (-x9 + 1)
[Step 7]  0 <= -x10 - x5 + 1 = (-x10 + 1) * (-x5 + 1)
[Step 8]  0 <= -x7 - x9 + 1 = (-x9 + 1) * (-x7 + 1)
[Step 9]  0 <= -x1 - x5 + 1 = (-x1 + 1) * (-x5 + 1)
[Step 10] 0 <= -x3 - x4 + 1 = (-x3 + 1) * (-x4 + 1)
[Step 11] 0 <= -x2 - x7 + 1 = (-x7 + 1) * (-x2 + 1)
[Step 12] 0 <= -x10 - x8 + 1 = (-x10 + 1) * (-x8 + 1)
[Step 13] 0 <= x3*x9 - x3 - x4 - x9 + 1 = [Step 2] * (-x3 + 1)
[Step 14] 0 <= x1*x3 - x1 - x2 - x3 + 1 = [Step 4] * (-x1 + 1)
[Step 15] 0 <= x4*x6 - x4 - x6 - x9 + 1 = [Step 1] * (-x4 + 1)
[Step 16] 0 <= -x6 + x8*x9 - x8 - x9 + 1 = [Step 0] * (-x9 + 1)
[Step 17] 0 <= x3*x5 - x3 - x4 - x5 + 1 = [Step 5] * (-x3 + 1)
[Step 18] 0 <= -x3 - x8 + 1 = (-x3 + 1)) * (-x8 + 1)
[Step 19] 0 <= x3*x7 + x3*x9 - x3 + x4*x7 - x4 - x7 - x9 + 1 = [Step 13] * (-x7 + 1)
[Step 20] 0 <= x10*x4 - x10 - x4 - x5 + 1 = [Step 5] * (-x10 + 1)
[Step 21] 0 <= -x10*x6 - x5*x6 + x6 = [Step 7] * (x6)
[Step 22] 0 <= x10*x9 - x10 + x5*x7 + x5*x9 - x5 - x7 - x9 + 1 = [Step 6] * (-x5 + 1)
[Step 23] 0 <= -x3*x7 - x7*x8 + x7 = [Step 18] * (x7)
[Step 24] 0 <= -x3*x7 - x3*x9 + x3 = [Step 8] * (x3)
[Step 25] 0 <= x4*x6 - x4 + x5*x6 + x5*x9 - x5 - x6 - x9 + 1 = [Step 15] * (-x5 + 1)
[Step 26] 0 <= -x1*x3 - x3*x5 + x3 = [Step 9] * (x3)
[Step 27] 0 <= x3*x6 + x3*x9 - x3 + x4*x6 - x4 - x6 - x9 + 1 = [Step 13] * (-x6 + 1)
[Step 28] 0 <= x3*x6 + x3*x9 - x3 - x6 + x8*x9 - x8 - x9 + 1 = [Step 16] * (-x3 + 1)
[Step 29] 0 <= x3*x6 - x3 - x6 - x8 + 1 = [Step 0] * (-x3 + 1)
[Step 30] 0 <= x1*x4 - x1 - x4 - x5 + 1 = [Step 5] * (-x1 + 1)
[Step 31] 0 <= x2*x4 + x2*x5 - x2 + x3*x5 - x3 - x4 - x5 + 1 = [Step 17] * (-x2 + 1)
[Step 32] 0 <= -x2 + x3*x7 - x3 - x7 + 1 = [Step 4] * (-x7 + 1)
[Step 33] 0 <= -x1*x4 - x2*x4 + x4 = [Step 14] * (x4)
[Step 34] 0 <= -x2*x5 - x5*x7 + x5 = [Step 11] * (x5)
[Step 35] 0 <= x1*x9 - x1 - x6 - x9 + 1 = [Step 1] * (-x1 + 1)
[Step 36] 0 <= -x10 + x7*x8 - x7 - x8 + 1 = [Step 3] * (-x8 + 1)
[Step 37] 0 <= -x10*x4 - x4*x7 + x4 = [Step 3] * (x4)
[Step 38] 0 <= x10*x6 - x10 - x6 - x8 + 1 = [Step 0] * (-x10 + 1)
[Step 39] 0 <= -x1*x9 - x5*x9 + x9 = [Step 9] * (x9)
[Step 40] 0 <= -x10*x9 - x8*x9 + x9 = [Step 12] * (x9)
[Step 41] 0 <= -x3*x6 - x4*x6 + x6 = [Step 10] * (x6)
0 <= 1/5 * [Step 19] + 1/5 * [Step 20] + 1/5 * [Step 27]
    + 1/5 * [Step 29] + 1/5 * [Step 33] + 1/5 * [Step 30]
    + 1/5 * [Step 36] + 3/5 * [Step 41] + 1/5 * [Step 23]
    + 1/5 * [Step 34] + 3/5 * [Step 35] + 2/5 * [Step 21]
    + 3/5 * [Step 24] + 1/5 * [Step 14] + 3/5 * [Step 32]
    + 1/5 * [Step 40] + 1/5 * [Step 37] + 1/5 * [Step 22]
    + 2/5 * [Step 25] + 1/5 * [Step 28] + 3/5 * [Step 39]
    + 2/5 * [Step 38] + 1/5 * [Step 31] + 1/5 * [Step 26] = 4 - \sum_{i=1}^{10} x_i.  □
```

Table 1: Examples of proofs generated by our agent. Axioms are shown in blue, and derived polynomials (intermediate lemmas) are shown in red. Note that the coefficients in the proofs are all *rational*, leading to exact and fully verifiable proofs. The first graph is the complete graph, the second graph is a randomly generated graph, and the third graph is the Petersen graph.