[Reviews · NeurIPS 2019]

Reviewer 1



The work seems to instantiate the general setting of combining ML/RL with proof guidance. The instantiation however seems quite different from existing theorem provers and applied to an area typically not considered by the core ATP community. Perhaps the discussion of related work could mention and compare with theorem proving in algebra with ATPs such as Otter, Prover9, Waldmeister and E, and the work on their ML-based guidance. It is far from correct that ML-based guidance focuses on tactical systems. Most of the work so far has been done on tableau system and saturation-style systems (leanCoP/rlCoP, E/ENIGMA) that learn guidance of elementary inference rules in their calculi. Some of the ideas could be explained in a more high-level way. For example, the intuition behind the reward function. The evaluation could also be made more convincing by including more problems than just stable set. --- Reply: I have read the reviews and the authors feedback. I will keep my score - the method and results are interesting and worth publishing. I have some additional questions/comments that could be taken into account in the final version: 1. The architecture of the neural net and alternative ML methods (raised also by Reviewer 3): 1a. Nonlinearity is not an automatic argument for neural nets. For example, gradient-boosted trees have been used successfully in a number of nonlinear and symbolic domains, also involving RL. 1b. Comparison/discussion of the custom architecture used here with graph neural nets (GNN) would be interesting. Also because GNNs have been used in neural SAT solving (cited here), and SAT could be in used to attack the stable set problem. 2. Would a neural-SAT/RL architecture be a viable alternative for stable set? 3. In general, more ablation studies showing the influence of the neural/RL hyperparameters would be useful. 4. ATP References: The common inspiration for the recent multitude of ML-guided tactical systems is Gauthier's TacticToe - http://arxiv.org/abs/1804.00596 .

Reviewer 2



As described above, I am quite positive about the paper. I consider the proposed dynamic proof technique as a valuable contribution for several problems in optimization and machine learning. The paper considers an optimization problem for which solvers are dominated by static approaches. Of course, for static approach, we may always construct examples for which the worst case run-time is attained. On the contrary, the dynamic approach can take a shortcut to the final result. The main drawback of the approach is the following doubt: Does the proposed dynamic approach always find a solution (e.g. a fully verifiable proof)? ------------------ I have read all reviews and the authors responses. The answers support the good quality of the paper. I recommend to accept it.

Reviewer 3



The authors tackle the general problem of searching for proofs of polynomial inequalities, which includes many computational problems of interest. They design a ML agent to search for dynamic semi-algebraic proofs; this appears to be the first successful attempt of its kind. The agent is a DQN whose neural network architecture handles polynomial inequalities in a way that takes into account the symmetries of the problem, and is trained in an unsupervised environment. The authors demonstrate its effectiveness on the maximum stable set problem (NP-hard in general), compared to existing static methods, and based on experiments, argue for the efficiency of searching for a dynamic rather than a static proof. A dynamic proof is also more interpretable, and more similar to a human-constructed proof. This is complementary to work on learning strategies for automated theorem proving. More precisely, the agent works as follows. It contains a memory of polynomials that are known to be nonnegative, and those known to be zero. At each time step $t$, the agent solves a LP to find the current best bound $\gamma_t$ certified using a linear combination of polynomials in the memory. It uses the DQN to select an action (multiply one of the polynomials in memory by $x_i$ or $1-x_i$). The reward is the relative improvement at each step, which requires no supervision. The architecture impose two symmetries: (1) that the Q-function is invariant to the order of polynomials in memory, and (2) that it is invariant to relabeling of variables. The authors make a large improvement over previous methods. Personally I find this line of work very promising. The paper also has a clear exposition on both the theoretical and engineering aspects, especially on explaining static vs. dynamic proofs. Questions and suggestions: + The maximum stable set problem is the only problem considered in the paper. Have the authors tried their method on other problems and found improvements? (The fact that static proofs of level l cannot certify bounds better than n/l seems to make this problem particularly favorable for showing the success of the dynamic method. What about other problems where there is no clear bound like this?) + How essential is it that the agent uses a neural network? How much of the improvement is due to using the reinforcement learning framework (perhaps with a simple classifier), vs. due to specifically using a neural network? + I would very much like the source code to be made available. --- Reply: Thanks for the authors' response. It is good to hear of promising results for other combinatorial problems despite the differences in distribution.

[Author Response · NeurIPS 2019]

We thank the reviewers for their positive feedback. Please find below a reply to the individual comments.

**Reviewer 1.**

*Perhaps the discussion of related work could mention and compare with theorem proving in algebra with ATPs [...]*

Thank you for the comment. We will correct this in the updated manuscript, and add references to papers that learn guidance of elementary inference rules.

*Some of the ideas could be explained in a more high-level way. For example, the intuition behind the reward function.*

Our choice of the reward function has a simple interpretation: a positive reward is given whenever the chosen action results in a bound improvement. That is, if the initial bound is $p(x) \leq 20$, and the chosen action has allowed to establish $p(x) \leq 19$, the agent is rewarded 1. This reward scheme has the benefit of being dense (as rewards are given gradually), in addition to being unsupervised (no ground truth bound required). The specifics of how we compute the bound at each step is described in Sec. 4.2. We will complement the current description in the paper with a more intuitive explanation in Sec. 4.2.

*The evaluation could also be made more convincing by including more problems than just stable set.*

Please see answer to Reviewer 3.

**Reviewer 2.**

*Are there situations for which [the dynamic approach] takes much longer than the static approach? If yes, is it possible to identify these situations?*

Thank you for raising this important point. In general, the dynamic approach cannot take much longer than the static approach; in fact, if we set the proof length of the dynamic method to be sufficiently large, all the proof generators (of the form $\mathbf{x}^\alpha (1 - \mathbf{x})^\beta$) will eventually be added to the memory, provided the degree of intermediate polynomials is not restricted. In this case, the LPs solved by the static approach and the dynamic approach are the same, hence overall the dynamic approach will only take slightly more time due to intermediate computations. We will add this point to the updated manuscript.

*Does the proposed dynamic approach always find a solution (e.g. a fully verifiable proof)? Could it happen that the approach fails completely?*

The completeness of the proof system guarantees that a proof of finite length exists for any positive polynomial. Hence, if we run the dynamic approach for long enough, a proof will be found. We note that in our experiment, we restrict the degree of the intermediate lemmas to 2. In this case, we cannot provably guarantee that the method will find the optimal bound all the time, however the bounds obtained in practice are much better those obtained by the static method.

**Reviewer 3.**

*Have the authors tried their method on other problems and found improvements?*

Thank you for raising this point. We first note that the stable set problem is sufficiently general in that many combinatorial problems (e.g., max matching, SAT, etc...) can be cast as specific instances of the stable set problem, via a simple reduction. We have done some preliminary experiments using our prover (trained on random graphs) to see how our method performs on such problems. Specifically, we evaluated the prover on the maximum matching problem on random graphs of size 30, where we used the property that the maximum matching corresponds to the maximum stable set of the associated line graph. Our results show that the proposed dynamic approach significantly outperforms the static approach, even though these problems come from a distribution that is different from what is seen during training. In particular, while the dynamic approach obtained an average upper bound of 13.8 with an LP of size $\leq 200$, the static approach only matched this upper bound by going to the 4th level of the hierarchy, with an LP of size $\approx 1.1 \times 10^6$. We therefore strongly believe our proposed approach can go well beyond the stable set problem, and can tackle other combinatorial problems via such reductions.

It should be noted that other (potentially more efficient and direct) ways of modeling such problems using polynomials also exist. While we did not try to train an agent specifically with these formulations, we see this as an exciting avenue of future research.

*How essential is it that the agent uses a neural network? How much of the improvement is due to using the reinforcement learning framework (perhaps with a simple classifier), vs. due to specifically using a neural network?*

In our setting, the neural network is used to decide the next proof step from the current proof state. To model this complex procedure, it is crucial to use a highly expressive and *nonlinear* function class. In addition, as argued in the paper, the Q-value function approximator has to incorporate important symmetries about the problem. It can be shown that if one uses a *linear* function, incorporating the "variable relabeling" symmetry significantly constrains the function and drastically reduces the degrees of freedom of the linear function. In contrast, neural networks have much larger expressivity, and were shown to be very successful value function approximators with good generalization properties in previous works.

[Meta-Review · NeurIPS 2019]

The reviewers agree that the contributions made in this submission are significant. Most of their questions and comments were successfully addressed in the rebuttal and they all recommended acceptance.